# Transcriptional Regulation of the *Phenylalanine Ammonia-Lyase* (*PAL*) Gene Family in Mulberry Under Chitosan-Induced Stress

**DOI:** 10.3390/plants14172783

**Published:** 2025-09-05

**Authors:** Apidet Rakpenthai, Mutsumi Watanabe, Arunee Wongkaew, Sutkhet Nakasathien

**Affiliations:** 1Department of Agronomy, Faculty of Agriculture, Kasetsart University, Bangkok 10900, Thailand; apidet.r@ku.th (A.R.); arunee.wo@ku.th (A.W.); 2Plant Secondary Metabolism, Division of Biological Sciences, Graduate School of Science and Technology, Nara Institute of Science and Technology, Nara 630-0192, Japan; mutsumi@bs.naist.jp

**Keywords:** *phenylalanine ammonia-lyase* (*PAL*), transcription factors (TFs), transcription factor binding site, mulberry, phenylpropanoid metabolism

## Abstract

Regulation of the phenylpropanoid pathway is critical for plant development and defense. This research investigates the transcriptional control of six *Phenylalanine Ammonia-Lyase* (*PAL*) gene homologs identified in the mulberry genome. A comprehensive in silico pipeline was employed to analyze the promoter architecture of these genes. Using the MEME suite, we identified three statistically significant conserved motifs within the 2000 bp upstream region. Subsequent TF binding prediction with FootprintDB for these motifs implicated the TCP, NAC, AP2/ERF, B3, and BBR-BPC families as potential regulators. A parallel analysis with PlantRegMap highlighted a high density of binding sites for the BBR-BPC and AP2/ERF families in the core promoter regions. A comparative analysis showed a weak correlation between the databases, underscoring the necessity of a multi-faceted predictive approach. Transcriptomic profiling under chitosan-induced conditions validated our in silico framework, suggesting the involvement of these TF families. Specifically, the data support NAC083 as a putative transcriptional activator and suggest a repressive function for members of the AP2/ERF and BBR-BPC families, providing a robust, experimentally supported model of *PAL* regulation.

## 1. Introduction

Phenylalanine ammonia-lyase (PAL) serves as a crucial gateway enzyme in the phenylpropanoid pathway of *Morus notabilis*, catalyzing the initial step of converting phenylalanine to *trans*-cinnamic acid. This pathway is fundamental for the biosynthesis of diverse bioactive compounds, including betulin, morusin, kuwanon G, sanggenon, chalcones, and flavonoids, which exhibit significant pharmacological properties such as anti-inflammatory, antioxidant, and anticancer activities [1]. The phenylpropanoid pathway in *M. notabilis* is particularly notable for producing unique prenylflavonoids and stilbenes that have garnered attention for their therapeutic potential [2]. Recent metabolomic studies have revealed that PAL activity directly influences the accumulation of these valuable secondary metabolites in mulberry tissues [3].

Multiple environmental and biotic factors can dramatically alter *PAL* expression, serving as critical regulatory mechanisms for phenylpropanoid pathway activation. Among abiotic stresses, drought stress significantly enhances *PAL* gene expression, with studies showing up to four-fold increases compared to control conditions [4]. Salt stress elicits more complex responses, where some *PAL* isoforms are upregulated while others may be downregulated, as observed in *Lotus japonicus*, where salt treatment altered expression of several *PAL* isogenes [5]. Other abiotic stimuli, such as UV radiation and mechanical wounding, are also well-documented inducers of *PAL*, further underscoring the enzyme’s central role in the plant’s integrated response to diverse environmental challenges. In addition to these abiotic triggers, a range of biotic elicitors strongly influence *PAL* activity. Pathogen-associated molecules, including fungal cell wall components and bacterial factors, rapidly activate *PAL* expression as part of the plant’s defense response [6]. Chitosan treatment, in particular, emerges as a highly effective elicitor for enhancing *PAL* expression, with its effectiveness demonstrated across multiple plant families, including the Moraceae. For instance, in *Ficus religiosa*, chitosan nanoparticle formulations have been developed and tested, though specific *PAL* expression studies remain limited [7], while in *Artocarpus heterophyllus*, chitosan treatment has been shown to modulate secondary metabolite production pathways [8]. This broad sensitivity to abiotic and biotic cues highlights *PAL* as a central hub in stress responses and suggests that the controlled application of such elicitors could serve as a promising biotechnological strategy to enhance secondary metabolite yields in agricultural systems.

Transcription factors (TFs) play pivotal roles in regulating the phenylpropanoid pathway across various plant species. MYB TFs, particularly R2R3-MYBs, have been extensively studied for their regulatory functions in flavonoid biosynthesis in Arabidopsis and other model plants [9]. The AP2/ERF TF family has been shown to modulate stress responses and secondary metabolism, with several members directly affecting phenylpropanoid pathway genes [10]. BBR-BPC TFs have emerged as important regulators of plant development and metabolism, influencing the expression of key biosynthetic genes [11]. Additionally, Dof TFs have been implicated in the regulation of phenylpropanoid metabolism, particularly in response to environmental stresses [12]. Recent studies have demonstrated that these TFs often work in complex regulatory networks, sometimes forming feedback loops with other regulatory elements to fine-tune metabolite production [13]. The intricate interplay between different TF families highlights the complex control mechanisms governing secondary metabolism in plants [14].

While some progress has been made in understanding transcriptional regulation in *M. notabilis*, particularly regarding silk protein production and stress responses [15], research on the specific regulatory mechanisms controlling phenylpropanoid pathway genes remains limited. Recent studies have identified several TFs involved in flavonoid biosynthesis in mulberry, but their precise roles in regulating PAL and other key pathway enzymes are not fully understood [10]. The complexity of mulberry’s secondary metabolism and the diverse array of bioactive compounds produced suggest the involvement of multiple regulatory networks that are yet to be fully elucidated [12]. This knowledge gap represents a significant barrier to the targeted genetic improvement of mulberry for enhanced phytochemical production. Uncovering these regulatory factors is crucial, as it could enable the engineering of *M. notabilis* to overproduce specific high-value compounds for pharmaceutical applications and simultaneously bolster its resilience to environmental stressors, thereby transforming it into an efficient biochemical factory that aligns with the goals of sustainable agriculture.

To illuminate the transcriptional control of *PAL* genes in *M. notabilis*, this study initiates a focused investigation using a suite of bioinformatics tools. We will systematically analyze the promoter regions of mulberry *PAL* genes to identify putative regulatory elements and the transcription factors that may interact with them. By correlating these in silico findings with the actual expression patterns of *PAL* and candidate TFs under chitosan treatment, we seek to uncover the key players driving phenylpropanoid biosynthesis. This work provides the foundational evidence for a complex regulatory system, proposing specific TFs as critical activators or repressors that guide metabolic flux in response to external stimuli.

## 2. Results

### 2.1. A Map of Grouped TFs via Known Cis-Elements in PAL Promoter Regions

The *PAL* gene family, crucial for phenylpropanoid metabolism, was found to have multiple homologous genes in mulberry through genome-wide identification. Our research identified 6 *PAL* genes in mulberry, expanding our understanding of this important enzyme family in a woody plant species. This analysis provides insights into the evolutionary history and potential functional diversity of *PAL* genes in mulberry.

To begin exploring the transcriptional regulation of *PAL* genes in mulberry, we utilized PlantRegMap to identify key TF families linked to known binding sites in mulberry. This tool, which leverages the FunTFBS algorithm, allowed us to explore the interaction between *cis*-elements and TFs, predicting transcription factor binding sites (TFBSs) by coupling TF binding affinities with evolutionary footprints across multiple plant species. The analysis is based on known motifs and annotated regulatory elements within the core promoter region, typically the conserved 200 bp upstream of the translation start site (ATG). The identified TF families included AP2/ERF, BBR-BPC, bHLH, C2H2, Dof, GATA, GRAS, HD-ZIP, MIKC_MADS, MYB, MYB-related, SBP, Trihelix, WOX, YABBY, and ZF-HD (Figure 1; Appendix A). Notably, the *PAL* promoters exhibited an enrichment of binding sites for AP2/ERF and BBR-BPC TF families.

A remarkable aspect of our findings is the high frequency of BBR-BPC and Dof TFs in certain *PAL* gene homologs (Figure 1). For instance, LOC21407112 shows 21 occurrences of BBR-BPC binding sites, and LOC21407114 has 17 occurrences of Dof binding sites. This repetitive nature suggests potential mechanisms for enhanced or modulated gene expression responses. BBR-BPC TFs have been shown to play multifaceted roles in plant development, stress responses, and immunity [16]. Dof TFs, on the other hand, are known to be involved in various plant-specific processes, including phenylpropanoid metabolism [17]. The high frequency of these binding sites in *PAL* gene homologs suggests a potential for enhanced or modulated gene expression responses, which could be crucial for regulating phenolic compound synthesis during stress responses or developmental processes in mulberry. This finding contributes to our understanding of the complex regulatory mechanisms governing secondary metabolism in plants and opens up new avenues for research in mulberry and other species.

The heatmap displays the predicted binding frequencies of various TF families to six *PAL* gene promoters (LOC21384641, LOC21407112, LOC21407113, LOC21407114, LOC21407115, and LOC21409963) in mulberry, as analyzed through PlantRegMap. Values in cells indicate the number of occurrences, with color-coding where green represents higher frequencies and yellow represents lower frequencies. Empty cells indicate the absence of the corresponding transcription factor family in that locus.

### 2.2. Identifying Conserved Motifs in the Promoter Regions of PAL Genes

In the analysis of promoter sequences from six *PAL* genes in *Morus* homologs, three conserved motifs (A, B, and C) were identified within 2000 bp upstream of the translation start site using MEME suite 5.1.1. These motifs, ranging from 10 to 12 nucleotides in length, all exhibited E-values of ≤0.05, indicating their statistical significance. The consensus sequences and positions of these motifs within the *PAL* gene promoters are illustrated in the provided Figure 2 and Table 1. Motif A, represented by a red color with a consensus sequence of TCCGACGATGCAGCMCAS, shows a highly conserved TCC-rich sequence appearing predominantly on the positive strand (−343 to −1922 bp, E-values as low as 2.73 × 10^−12^) in all six analyzed *PAL* genes (LOC21384641, LOC21407112, LOC21407113, LOC21407114, LOC21407115, and LOC21409963). Motif B, depicted in blue, has a CAG-rich consensus sequence (CAGCCAAATCAMAGCM) present in five genes with positions varying from −32 to −1824 bp, while Motif C, shown in green, reveals a CT-rich pattern (CTMTCNCTCTCTCTAGCY) located between −71 and −816 bp in all six *PAL* genes. All three motifs demonstrate strong statistical significance across the sequences, with notably high levels of sequence preservation specifically observed in the promoter regions of LOC21407112 and LOC21407113, suggesting their potential role as important regulatory elements in these promoter regions.

Comprehensive depiction of promoter regions in mulberry homolog genes for *PAL*. The promoter regions shown span 2000 base pairs upstream of the translation start site (ATG). The diagrams highlight motifs A-C, consensus regions predicted from MEME, across various gene sequences. Three motif types are depicted using different colors: red (Motif A, sequence TCCCACCATGCACCMCAS), blue (Motif B, sequence CAGCCAAATCAMAGCW), and green (Motif C, sequence CTWTCNCTCTCTCTAGCY). The minus (−) symbol signifies that the motif is situated on the anti-sense strand of DNA.

The table is organized into columns representing three different motifs (A, B, and C), with each motif shown in both sequence logo format and nucleotide sequence. The sequence logos at the top of each motif column visualize the consensus sequences using colored letters, where the height of each letter represents its relative frequency at that position. These sequences include specific IUPAC-coded bases to represent nucleotide variations: ‘M’ (A/C), ‘N’ (A/C/G/T), ‘S’ (G/C), and ‘Y’ (T/C). For each identified motif, the analysis provides detailed information including the exact matching sequence in the *PAL* promoters, its position, strand orientation (either positive or negative), and the statistical significance of sequence similarity expressed as E-values from Blastp comparisons.

### 2.3. In Silico Exploration for TF Candidates in PAL Promoter Conserved Motifs

As a meta-database that integrates and curates high-quality, experimentally validated TFBS data from a multitude of primary public repositories such as JASPAR and UniPROBE, FootprintDB has demonstrated remarkable accuracy in predicting both plant and animal transcription factors (TFs) that interact with specific *cis*-regulatory sequences, showing strong correlation with experimental yeast one-hybrid (Y1H) results [18]. It particularly focuses on high-confidence, experimentally supported TFBS predictions, especially for known DNA-binding motifs. In the study by Rakpenthai et al. (2022) [19], FootprintDB successfully identified 14 candidate TFs involved in regulating *SDI* gene expression, including key factors like SLIM1, HYH, and GBF1, which were later experimentally validated through Y1H screens and electrophoretic mobility shift assays. Building on these successful applications, this powerful tool for predicting TFBSs was employed to identify potential TFs that may bind to motifs A-C in the promoters of six mulberry *PAL* promoters (Figure 3a; Appendix A). For motif A, the analysis predicted binding sites for several TFs from the TCP family, including TCP1, TCP21, TCP22, AT2G45680, AT5G08330, AT5G51910, TCP19, TCP20, TCP7, and TCP9, all belonging to the TCP TF family. Motif B was associated with a diverse group of TFs, including ERF096, ERF1, ERF2, ERF5, and ERF1B from the AP2/ERF family, as well as DOF2.2 from the Dof family, and several NAC family members such as NAC060, NAC058, NAC046, and ANAC100. For motif C, the analysis identified potential binding sites for TFs from the AP2/ERF family (RAV2, TEM2, TEM1, EDF3), the B3 family (REM1), and the BBR-BPC family (BPC5, BPC3, BPC1). Additionally, FRS9 from the FRS family was also associated with motif C, while DOF2.2 from the Dof family was linked to Motif B. The identification of these less commonly studied families, alongside the well-established AP2/ERF and NAC TFs, suggests a multi-layered regulatory network. The involvement of Dof TFs is particularly interesting given their known roles in phenylpropanoid metabolism, and the presence of an FRS binding site points to potentially novel regulatory mechanisms governing *PAL* gene expression in mulberry. The Venn diagram illustrates the distribution of TFs across six mulberry *PAL* gene promoters: LOC21384641, LOC21409963, LOC21407112, LOC24107113, LOC24107114, and LOC24107115 (Figure 3b; Appendix A). The distribution pattern reveals a high degree of specificity among the TFs, with LOC21384641 having the highest number of unique TFs (24, 50%), followed by LOC21409963 with 9 (19%) unique TFs. Interestingly, there are no TFs shared by all six promoters simultaneously, indicating a diverse regulatory landscape for these *PAL* genes.

### 2.4. Correlation Study of TFBS Predictions by PlantRegMap and FootprintDB in PAL Promoters

The scatter plot compares the counts of TF families between PlantRegMap and FootprintDB databases, potentially relating to the *PAL* gene promoter (Figure 4; Appendix A). There appears to be a weak positive correlation (R^2^ = 0.043) between the two databases, as indicated by the upward-sloping red trend line. The AP2/ERF family stands out with the highest count in FootprintDB (10) and a moderate count in PlantRegMap (27), suggesting it may be a significant regulator of the *PAL* gene. Other notable TF families include TCP and NAC, which show high FootprintDB counts but lower PlantRegMap counts. The BRR_BPC family, conversely, has a high PlantRegMap count (62) but a lower FootprintDB count. This differential representation suggests that AP2/ERF binding sites are more consistently recognized across both databases, while BBR-BPC binding sites show database-specific recognition patterns. This distinct distribution pattern suggests that both families likely play different but complementary roles in *PAL* promoter regulation, with AP2/ERF showing consistent recognition across databases and BBR-BPC displaying database-specific prominence, particularly in PlantRegMap. This discrepancy in counts between databases for various TF families highlights the importance of considering multiple data sources when studying potential regulators of the *PAL* gene promoter, as each database may capture different aspects of TF binding potential. The weak correlation is an expected outcome, reflecting the orthogonal nature of the two platforms: PlantRegMap’s discovery-based approach relies on evolutionary conservation, while FootprintDB’s annotation-based method is constrained by the scope of existing experimental data. Using both tools provides a more robust analysis by combining broad, evolutionarily informed hypotheses with targeted, evidence-based predictions.

Scatter plot illustrating the comparative analysis of TF family counts between two databases. The x-axis represents counts from PlantRegMap (ranging from 0 to 60), while the y-axis shows counts from FootprintDB (ranging from 0 to 10). Each point on the plot represents a different TF family labeled with abbreviations such as TCP, NAC, B3, MYB, and others. A red regression line is drawn through the data points, and there is an R-squared value (R^2^ = 0.043).

### 2.5. Transcriptional Response of Candidate Regulators of PALs Under Conditions That Alter PAL Expression

Given that chitosan effectively triggers *PAL* expression across diverse plant families and has demonstrated particular efficacy in species closely related to mulberry within the Moraceae family [7,8], chitosan treatment in mulberry can similarly be expected to alter *PAL* expression. RNA sequencing analysis from mulberry root revealed a significant upregulation of *PAL* gene expression in samples treated with chitosan compared to untreated controls, confirming the elicitor role of chitosan in activating plant defense pathways (Figure 5; Appendix A). Consistent with this, the *pathogenesis-related protein 1* (*PR1*) gene, a known marker for induced defense responses, also exhibited elevated transcript levels in the chitosan-treated condition. Among the candidate transcription factors analyzed, *NAC083* showed a clear upregulation, suggesting its potential role in positively regulating *PAL* expression. In contrast, *ERF2*, *ERF5*, *RAV2*, *HAT3*, *BPC6*, *REF6*, *CDC5*, and *TCP20* were significantly downregulated, indicating a possible repressive role or indirect regulatory relationship. Expression levels of *TCP7*, *TCP9* and *TCP19* remained unchanged, similar to the expression of the *actin housekeeping* gene, suggesting they may not be responsive to chitosan-induced *PAL* regulation. The remaining candidate TFs listed in Figure 3a were not detected in the dataset.

The bar chart displays the transcriptional response of selected genes in mulberry following chitosan treatment as compared to the FN condition, quantified as LOG_2_ Fold Change (LOG_2_ FC). The analyzed genes are grouped by color: blue bars represent candidate TF genes identified via in silico analysis; orange bars represent *Phenylalanine ammonia-lyase* (*PAL*) genes; the green bar represents a *Pathogenesis-related* (*PR*) gene, a positive control for chitosan induction; and the white bar represents the *actin housekeeping* gene. Positive LOG_2_ FC values indicate upregulation, while negative values indicate downregulation. Error bars indicate the standard error of the mean (SEM) from three biological replicates.

## 3. Discussion

### 3.1. A Complex Regulatory Network Involving AP2/ERF, BBR-BPC, and Other TF Families Controls PAL Gene Expression

A comprehensive bioinformatics analysis of TFBS for *PAL* gene promoters, utilizing both FootprintDB and PlantRegMap databases, has revealed two significant regulatory families: AP2/ERF and BBR-BPC. The AP2/ERF family demonstrates substantial presence in PlantRegMap with 27 binding sites distributed across 5 *PAL* gene core promoters (LOC21384641, LOC21407112, LOC21407113, LOC21407114, LOC21407115), indicating its crucial role in *PAL* gene regulation. Most notably, the BBR-BPC family emerges as the dominant regulator with an impressive 62 binding sites across all 6 *PAL* genes in PlantRegMap, with particularly high concentrations in LOC21407112 (21 sites), LOC21407113 (16 sites), and LOC21407115 (18 sites). The significance of BBR-BPC is further reinforced by its presence in FootprintDB’s motif C for 3 LOC, and its clear dominance is thus visualized in scatter plots comparing FootprintDB and PlantRegMap counts across TF families. Previous studies have shown that AP2/ERF TFs regulate phenylpropanoid metabolism and lignin biosynthesis [20], and BBR-BPC proteins serve as key regulators of various metabolic pathways [21]. The extensive presence of these TFBS collectively indicates a complex, multilayered regulatory network controlling phenylpropanoid pathway flux, which plays a crucial role in plant stress responses and development [22]. Beyond the prominent AP2/ERF and BBR-BPC families, our in silico analysis also pointed to other potentially significant regulators. For instance, the Dof family, with a high concentration of binding sites in the LOC21407114 promoter and a predicted interaction with Motif B, has been previously implicated in phenylpropanoid metabolism. Furthermore, the identification of an FRS family member binding to Motif C introduces another layer of complexity, suggesting that novel or less-characterized TFs could also be integral to fine-tuning *PAL* expression. While our transcriptomic data did not capture the expression of these specific TFs, their predicted binding sites warrant future investigation to build a more complete model of *PAL* regulation. The transcriptomic analysis in mulberry roots under chitosan treatment revealed a coordinated downregulation of transcription factors (TFs) belonging to the AP2/ERF (*ERF2*, *ERF5*, *RAV2*) and BBR-BPC (*BPC6*) families, concurrent with the upregulation of *PAL* transcripts (Figure 5). This inverse correlation suggests a potential repressive role of these TFs on *PAL* gene expression. The consistent reduction in expression across distinct TF families may indicate that they function either independently as transcriptional repressors or collaboratively as part of a multi-protein repressor complex that regulates *PAL* expression. These findings provide insight into the possible regulatory network involved in phenylpropanoid pathway activation under elicitor treatment and highlight the need for further functional characterization of these candidate TFs to confirm their repressive roles. Indeed, the modern understanding of the *cis*-regulatory code dictates that the function of such a network is governed not just by the presence of TFBSs, but by a complex syntax where TF activity is modulated by binding site position, strand orientation, and the local chromatin context [23,24].

### 3.2. NAC083 Acts as an Activator in PAL Expression in Chitosan Application

In this study (Figure 5), the significant and parallel upregulation of both *NAC083* and *PAL* genes in response to chitosan treatment strongly implicates NAC083 as a putative transcriptional activator in the mulberry phenylpropanoid pathway. Our observation of a coordinated four-fold increase in their expression levels provides compelling evidence for a functional regulatory link. This hypothesis is substantiated by convergent findings in other plant species, suggesting a conserved evolutionary mechanism. For instance, functional characterization in Arabidopsis revealed that the overexpression of an *NAC083* ortholog directly induced the expression of key phenylpropanoid and flavonoid biosynthesis genes, including PAL [25]. This supports a direct or indirect role for NAC083 in modulating the pathway’s entry point. Furthermore, the potential for direct regulation is underscored by genomic analyses in soybean, which identified a prevalence of NAC transcription factor binding sites within the promoter regions of *PAL* gene families [26]. This indicates that the NAC-PAL regulatory module is a common feature in plant defense and secondary metabolism. The work by [25] also linked high *AeNAC83* expression to enhanced abiotic stress tolerance, which is likely conferred by the accumulation of protective phenylpropanoid-derived compounds. Therefore, we propose a model wherein chitosan, acting as an elicitor, triggers the expression of *NAC083*. This transcription factor then activates *PAL* gene expression, initiating a cascade of secondary metabolite production that fortifies the plant’s cellular defense system and mitigates stress. NAC083 is therefore a strong candidate as a key transcriptional regulator activating *PAL* expression under chitosan treatment in mulberry.

## 4. Materials and Methods

### 4.1. Identifying Known Binding Sites and Conserved Sequence Motifs in Mulberry PAL Gene Promoters

The promoter sequences of the *PAL* genes in mulberry, specifically *Morus notabilis*, were comprehensively determined using KEGG (Kyoto Encyclopedia of Genes and Genomes) accessible at (https://www.kegg.jp/kegg-bin/show_organism?org=mnt (accessed on 1 February 2025); [27]). This analysis involved sequencing about 2000 base pairs upstream from the translation start site to pinpoint potential regulatory elements (Appendix A). Additionally, TF families and their respective binding sites on mulberry *PAL* genes were identified through the PlantRegMap database (https://plantregmap.gao-lab.org/ (accessed on 1 February 2025); [28]), which focused its analysis on the core promoter region (200 bp upstream from ATG) to leverage evolutionary conservation for identifying key TFBSs. The MEME (Multiple EM for Motif Elicitation) suite v5.1.1 was employed to analyze the 2000 bp region upstream of the *PAL* genes. This analysis aimed to identify key regulatory DNA motifs by capturing both proximal and distal regulatory elements. Using the online tool (http://meme-suite.org/tools/meme (accessed on 1 February 2025); [29]), the analysis was configured to detect up to 20 distinct motifs, each 16–18 nucleotides long, typical of transcription factor binding sites. Motifs were filtered using an E-value threshold of ≤0.05 to ensure statistical significance based on log likelihood ratio, motif length, and frequency of occurrence.

### 4.2. Assessment of Candidate TFs Regulating PAL via Motifs and Database Correlation Analysis

To identify potential TFs regulating *PAL* genes, conserved sequence motifs previously identified by MEME in promoter regions were analyzed using FootprintDB (http://floresta.eead.csic.es/footprintdb (accessed on 15 February 2025); [30,31]). This targeted approach was chosen to enhance prediction accuracy by focusing on regions with demonstrated statistical significance. FootprintDB integrates curated DNA binding sites and annotates TF binding interfaces, and using a strict E-value threshold of 10^−3^ for Blastp alignments against the 3D-footprint library ensured high-confidence predictions. The predicted TFs for each *PAL* gene (LOC) were compiled into a table list, and their distribution and overlap were visualized using a Venn diagram created with the ggVennDiagram package [32] in RStudio software (Version 2023.09.1, Build 494) [33]. Subsequently, correlation analysis between TF families from PlantRegMap and FootprintDB was performed using R (version 4.3.2; [34]) and RStudio. The analysis involved two datasets: TF family frequencies from PlantRegMap’s core promoter analysis across six *PAL* genes, and motif counts from FootprintDB. Data preprocessing included summing TF occurrences across *PAL* genes for PlantRegMap data and FootprintDB. Missing values were replaced with zeros. Pearson’s correlation coefficient was calculated and visualized using the corrplot package to generate a color-coded correlation matrix, enabling quantitative assessment of TF family distributions.

### 4.3. Plant Sample Collection and RNA Extraction

Morus ‘Sakonnakorn’ stem cuttings from one-year-old plants were cultivated in a greenhouse and an aeroponic system for 2.5 months under a 12 h light/12 h dark photoperiod and full-nutrient (FN) conditions maintained at pH 6.5, with the nutrient solution containing the following compounds at specified final concentrations (all chemicals were sourced from Kemaus company): 2 mM KNO_3_, 2 mM Ca(NO_3_)_2_, 1 mM MgSO_4_, 0.67 mM KH_2_PO_4_, 50 μM NaCl, 50 μM C_10_H_12_N_2_NAFeO_8_, 5 μM MnSO_4_, 5 μM CuSO_4_, 5 μM ZnSO_4_, 165 μM H_3_BO_3_, 1 μM Na_2_MoO_4_, 5 μM CoSO_4_, and 5 μM NiSO_4_. The cuttings were treated for two weeks in FN medium supplemented with medium molecular weight chitosan at a final concentration of 0.04% (*w*/*v*). This was achieved by dissolving 16 g of chitosan in 40 L of the medium and adjusting the final pH to 6.5 with NaOH. After treatment, the entire roots of both treated and untreated cuttings were rinsed with deionized water and subsequently harvested by freezing in liquid nitrogen. 50 mg ground root samples were used for Total RNA extraction done by GF-1 Total RNA Extraction Kit (Vivantis Technologies Sdn. Bhd., Cat. No. GF-TR-025). The quality of the extracted RNA was checked using a FastQC analysis to assess the Phred quality score at each cycle.

### 4.4. cDNA Synthesis and Preparation of the Library for High-Throughput Sequencing

The mulberry RNA samples were sent to Macrogen, Inc. (Seoul, Republic of Korea) for sequencing, where mRNA was enriched using oligo(dT) magnetic beads and subsequently fragmented to an average length of 151 bp. First-strand cDNA synthesis was performed using random primers, followed by second-strand synthesis with DNA Polymerase I and RNase H, incorporating dUTP to quench the second strand during amplification. Double-stranded cDNA was purified using the TruSeq Stranded mRNA Library Prep Kit (Illumina, San Diego, CA, USA; Cat. No. RS-122-2101) and washed with ELB buffer. Sequencing adaptors were ligated to the cDNA fragments, which were then enriched by PCR with a primer cocktail. The size of the PCR-enriched fragments was verified by running the samples on an Agilent Technologies 2100 Bioanalyzer using a DNA 1000 chip. The prepared libraries were quantified using qPCR according to the Illumina qPCR Quantification Protocol Guide. The final library concentrations were required to be at least 10 nM for sequencing. The final libraries were prepared for sequencing on the Illumina HiSeq. 2000 instrument.

### 4.5. De Novo Transcriptome Assembly and Identifying Expression Differences

High–quality reads were filtered by eliminating low–quality reads with Q < 20 from the raw reads. To perform quality trimming and removal of adapters, the fast and multi–threaded command line tool ‘trimmomatic’ was used (http//www.usadellab.org/cms/?page=trimmomatic (accessed on 15 April 2025)). The resulting high-quality reads were aligned to the *Morus notabilis* reference genome (ASM41409v2) using HISAT2 (v2.1.0). StringTie (v2.1.3b) was then used for reference-guided transcript assembly and to quantify expression as raw counts, FPKM, and TPM. Significant alterations in the expression of candidate transcription factor (TF) genes were identified by calculating the log_2_-fold change from three biological replicates of the treatment group compared to the control. Standard error was determined using the method established by [35].

## 5. Conclusions

Our integrated bioinformatics and transcriptomic analyses have elucidated a complex regulatory network regulating the six identified mulberry *PAL* genes crucial for the phenylpropanoid pathway. We identified an enrichment of binding sites for the AP2/ERF and BBR-BPC transcription factor families within the *PAL* promoter regions. Transcriptional profiling following chitosan treatment, which significantly upregulated *PAL* expression, revealed a concomitant downregulation of several AP2/ERF (ERF2, ERF5, and RAV2) and BBR-BPC (BPC6) members, suggesting their potential role as transcriptional repressors. Conversely, the significant upregulation of NAC083 in parallel with *PAL* genes strongly implicates it as a key transcriptional activator in this response. These findings suggest a regulatory model where a balance between repressors and activators fine-tunes *PAL* expression. This research not only enhances our understanding of the transcriptional control of secondary metabolism in mulberry but also identifies promising candidate genes for future functional characterization and potential molecular breeding applications.

## Figures and Tables

**Figure 1 plants-14-02783-f001:**
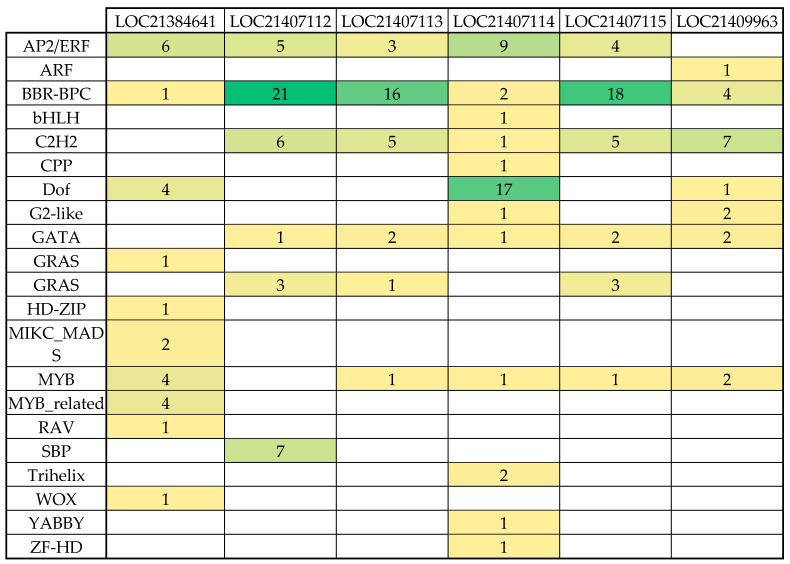
Distribution of transcription factor binding sites across *PAL* gene promoters in *M. notabilis*.

**Figure 2 plants-14-02783-f002:**
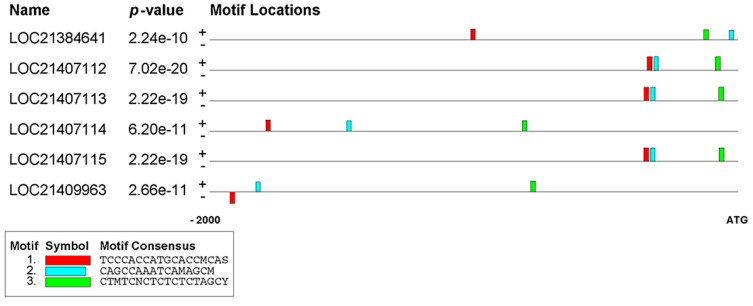
Motif distribution analysis in *PAL* promoters.

**Figure 3 plants-14-02783-f003:**
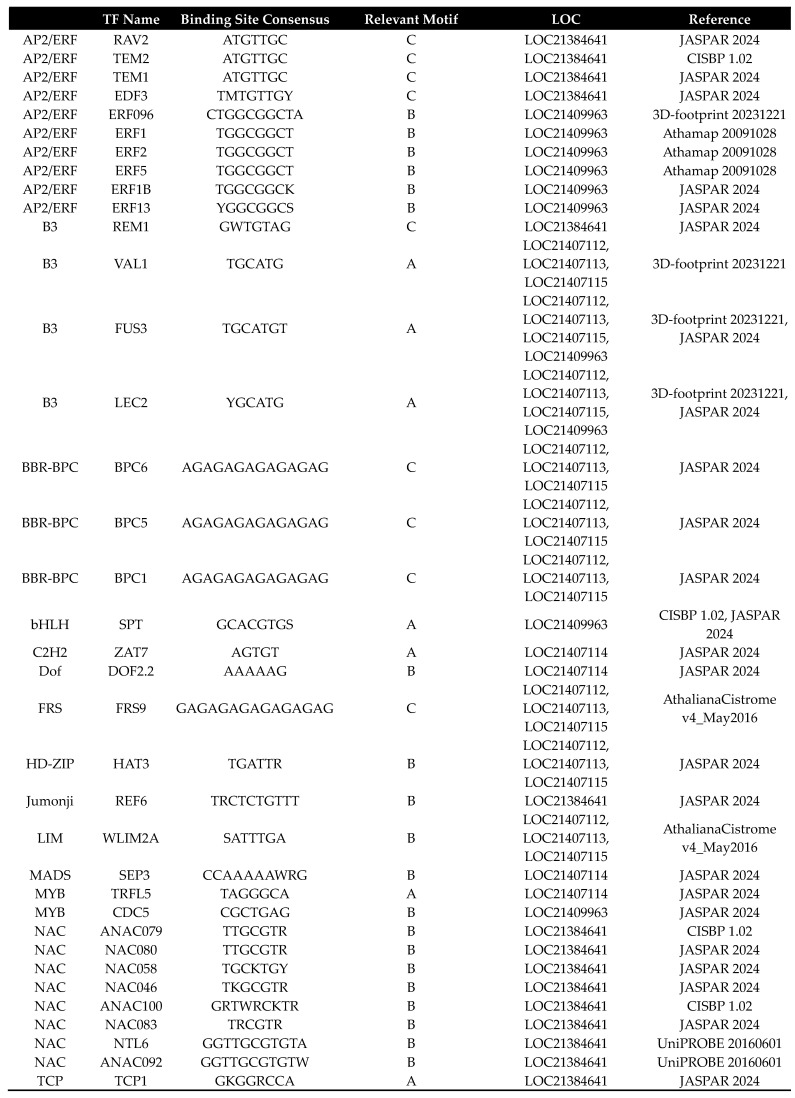
*Cis*-elements identified by FootprintDB in three conserved motifs in *PAL* genes. (**a**) The FootprintDB-identified *cis*-elements in three motifs across six *PAL* promoter regions, presenting them as paired combinations of binding sites (BS) and TF families, sorted alphabetically by the TF family name. Ambiguous nucleotide bases within the BS sequences are represented using IUPAC codes: R for A or G, Y for C or T, S for G or C, W for A or T, K for G or T, and M for A or C. The positions of all *cis*-elements are indicated relative to their location within the identified motif found in *PAL* genes, and references for each identified *cis*-element are provided. (**b**) The Venn diagram effectively illustrates how cis-elements identified by FootprintDB are distributed across six PAL promoter regions, utilizing numerical values and percentages to demonstrate both distinct and shared patterns. The diagram employs a three-tone color scheme (white, light blue, and dark blue), where white represents the lowest concentration of cis-elements, light blue indicates intermediate levels, and dark blue signifies the highest concentration of cis-elements.

**Figure 4 plants-14-02783-f004:**
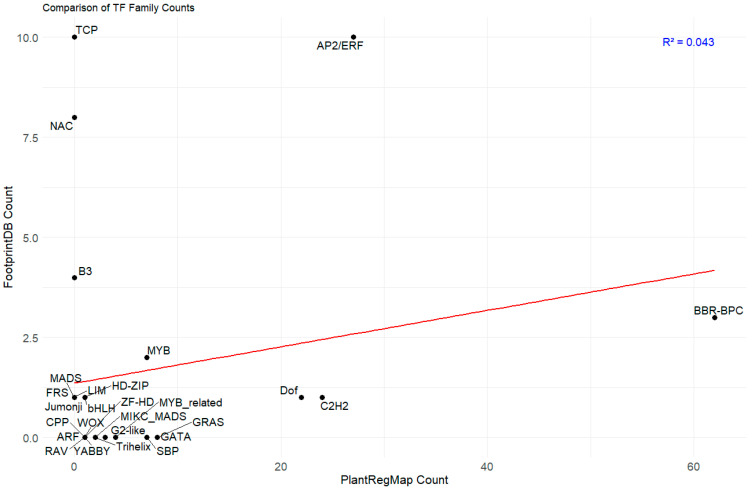
Cross-database TF family correlation analysis.

**Figure 5 plants-14-02783-f005:**
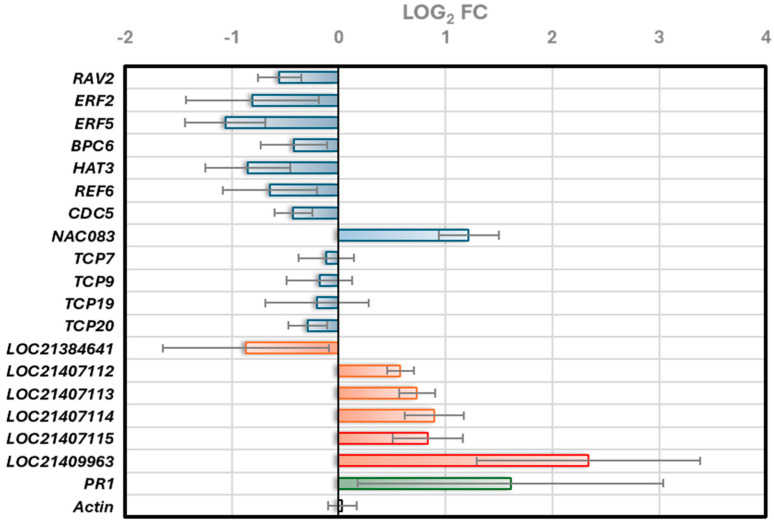
Transcriptional response of candidate transcription factors and *PAL* genes in chitosan-treated mulberry.

**Table 1 plants-14-02783-t001:** The MEME analysis identified three distinct motifs (A, B, and C) across six LOC sequences, revealing significant patterns of DNA conservation.

	**Motif**	**A**	**B**	**C**
	Motif Seq Logo	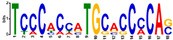 TCCCACCATGCACCMCAS	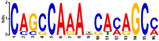 CAGCCAAATCAMAGCM	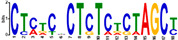 CTMTCNCTCTCTCTAGCY
LOC21384641	Matched	TCCCTCTGTGGGGCCCAC (+)	CACGCAAACCAAAGCA (+)	CTAGCACTCTATGTAGCT (+)
	Location	−1012	−32	−129
	E-value	9.54 × 10^−9^	5.06 × 10^−8^	1.73 × 10^−8^
LOC21407112	Matched	TCCCACCATGCACCCCAG (+)	CAGCCAAATCACAGCC (+)	CTCTCCCTCTCTCTAGCT (+)
	Location	−343	−317	−85
	E-value	2.73 × 10^−12^	7.20 × 10^−11^	4.17 × 10^−12^
LOC21407113	Matched	TCCCACCATGCACCCCAG (+)	CAGCCAAATCACAGCC (+)	CTCTCTCTCTCTCTAGCT (+)
	Location	−355	−329	−72
	E-value	2.73 × 10^−12^	7.20 × 10^−11^	1.38 × 10^−11^
LOC21407114	Matched	TCACACACTGCCCCACAC (+)	CAGCCAAAAGAAGGCC (+)	CCAACCCTGTCACTAGCC (+)
	Location	−1787	−1480	−816
	E-value	5.09 × 10^−9^	2.74 × 10^−8^	1.52 × 10^−8^
LOC21407115	Matched	TCCCACCATGCACCCCAG (+)	CAGCCAAATCACAGCC (+)	CTCTCTCTCTCTCTAGCT (+)
	Location	−356	−330	−71
	E-value	2.73 × 10^−12^	7.20 × 10^−11^	1.38 × 10^−11^
LOC21409963	Matched	TTCCCACATGCACCACAC (−)	CCGCCAGAGCTCAGCA (+)	CTCTTGCTCTCGCAAGCC (+)
	Location	−1922	−1824	−784
	E-value	3.63 × 10^−9^	3.53 × 10^−8^	6.66 × 10^−9^

## Data Availability

The RNA-seq data from this study is publicly available in the NCBI Gene Expression Omnibus (GEO), with the accession number GSE303913. For access to other datasets from this study, please contact the corresponding author.

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
