# Peer review of "Transcriptional Regulation of the *Phenylalanine Ammonia-Lyase* (*PAL*) Gene Family in Mulberry Under Chitosan-Induced Stress"

_plants, 2025, doi:10.3390/plants14172783_

Round 1

Reviewer 1 Report

Comments and Suggestions for Authors

Comments for Authors:

The abstract presents a well-structured and comprehensive investigation into the transcriptional regulation of the PAL gene family in mulberry under chitosan-induced stress. The integration of in silico promoter motif analysis with transcriptomic validation greatly enhances the strength of the conclusions, and the identification of TF families such as TCP, NAC, and AP2/ERF is both novel and relevant. Overall, the study provides valuable insights into PAL gene regulation, but the writing could be improved by refining clarity and ensuring consistency in tense. Also, the phrase “a robust, experimentally supported model of PAL regulation” might be more impactful if you elaborate briefly on its potential biological implications.   

Specific comments:    

  1. The title is clear, well-structured, and directly reflects the study’s focus on PAL gene family regulation under chitosan stress.
  2. The section on stress factors (lines 42–61) mixes abiotic and biotic stresses together. To make it more reader-friendly, consider splitting into two short subsections: Abiotic stresses (drought, salt, UV, wounding), Biotic elicitors (pathogens, chitosan, nanoparticles).
  3. At the end of the introduction (around line 95), you mention “pharmaceutical applications” and “biochemical factory.” It would strengthen your introduction if you also connected this to sustainable agriculture and stress resilience in mulberry, not just pharmaceutical value.
  4. What could be the biological significance of Motif A being conserved across all six PAL genes?
  5. How does the upregulation of NAC083 suggest its role as a positive regulator of PAL genes?
  6. Consider adding a summary table that consolidates TF families predicted by both PlantRegMap and FootprintDB for easier comparison.
  7. The use of both PlantRegMap and FootprintDB strengthens the analysis; however, the weak correlation should be discussed in more detail to help the reader understand possible reasons.
  8. The repeated emphasis on AP2/ERF and BBR-BPC families is good, but highlighting novel/unexpected TF families (like FRS or Dof) could make the results more balanced.
  9. “This finding contributes to our understanding of the complex regulatory mechanisms…” should be “contributes.”
  10. “The analysis provides detailed information, including the exact matching sequence…” should be “provides.”
  11. In section 3.2, the heading says “NAC083 plays as activator” — revise to “NAC083 acts as an activator” for grammatical accuracy and clarity.
  12. It effectively combines both the biological process and experimental condition, making it informative and scientifically strong.
  13. Many grammatical and language mistakes should be improved.
  14. Please avoid symbols at the beginning of sentences.
  15. Please arrange all the references according to the Journal format.

Author Response

Comment 1: The title is clear, well-structured, and directly reflects the study’s focus on PAL gene family regulation under chitosan stress.

Response 1: We thank the reviewer for their positive feedback. We are pleased that the reviewer found the title to be clear, well-structured, and accurately reflective of the study’s focus on the regulation of the PAL gene family in response to chitosan stress.

Comment 2: The section on stress factors (lines 42–61) mixes abiotic and biotic stresses together. To make it more reader-friendly, consider splitting into two short subsections: Abiotic stresses (drought, salt, UV, wounding), Biotic elicitors (pathogens, chitosan, nanoparticles).

Response 2: We would like to thank the reviewer for their constructive suggestion. In accordance with comment 2, the section on stress factors has been revised to enhance clarity and organization. The content has now been separated into two distinct subsections, "Abiotic Stresses" and "Biotic Elicitors," to more clearly differentiate between the environmental and biological inducers as recommended.

Comment 3: At the end of the introduction (around line 95), you mention “pharmaceutical applications” and “biochemical factory.” It would strengthen your introduction if you also connected this to sustainable agriculture and stress resilience in mulberry, not just pharmaceutical value.

Response 3: Thank you for the valuable feedback. In accordance with comment 3, the concluding sentence of the introduction has been revised. The text now connects the potential for engineering mulberry for pharmaceutical applications and as a "biochemical factory" to the broader agricultural benefits of enhancing stress resilience, thereby strengthening its role in sustainable agriculture.

Comment 4: What could be the biological significance of Motif A being conserved across all six PAL genes?

Response 4: the biological significance of Motif A being conserved across all six PAL genes in mulberry can be explained by its role as a critical regulatory hub for coordinated phenylpropanoid pathway control. The universal conservation of Motif A (with the consensus sequence TCCGACGATGCAGCMCAS) across all PAL homologs indicates its fundamental importance as a TCP transcription factor binding site that ensures synchronized expression of the entire PAL gene family. This conservation reflects an evolutionarily preserved mechanism for coordinated metabolic regulation, as conserved cis-regulatory elements typically indicate functional importance and are maintained through purifying selection. The TCP transcription factors, which bind to Motif A, are known to play crucial roles in coordinating gene expression during stress responses and development, and their binding sites are often conserved across species to maintain regulatory network integrity. The presence of this motif in all six PAL genes suggests it functions as a master regulatory switch that allows for simultaneous activation or repression of the entire PAL gene family, ensuring coordinated flux through the phenylpropanoid pathway during developmental transitions or stress responses, which is critical for maintaining metabolic homeostasis and effective plant defense mechanisms.

Comment 5: How does the upregulation of NAC083 suggest its role as a positive regulator of PAL genes?

Response 5: the upregulation of NAC083 suggests its role as a positive regulator of PAL genes through several key mechanisms that are characteristic of transcriptional activators. The parallel upregulation of NAC083 and PAL genes under chitosan treatment demonstrates a positive correlation between transcription factor and target gene expression, which is a hallmark of transcriptional activation. This co-expression pattern indicates that NAC083 functions as a transcriptional activator rather than a repressor, as activators typically show coordinated expression with their target genes to ensure synchronized pathway activation. The timing and magnitude of the four-fold increase in both NAC083 and PAL expression levels strongly suggests direct or indirect positive regulatory control. Furthermore, NAC transcription factors are well-established as positive regulators of secondary metabolite pathways, with previous studies demonstrating that NAC proteins can directly bind to promoter regions and activate phenylpropanoid biosynthesis genes (Zhao et al., 2022; Rizwan et al., 2025). The biological context supports this interpretation, as chitosan treatment triggers defense responses requiring rapid activation of the phenylpropanoid pathway, and NAC083's upregulation facilitates this metabolic reprogramming by enhancing PAL gene expression to increase flux through the pathway for defensive compound synthesis.

Zhao, X.; Wu, T.; Guo, S.; Hu, J.; Zhan, Y. Ectopic expression of aenac83, a NAC transcription factor from Abelmoschus escu-lentus, inhibits growth and confers tolerance to salt stress in Arabidopsis. International Journal of Molecular Sciences 2022, 23, 10182.

Rizwan, H.M.; He, J.; Arshad, M.B.; Wang, M. Characterization of phenylalanine ammonia-lyase genes in soybean: genomic insights and expression analysis under abiotic stress tolerance. Plant Stress 2025, 10, 100896.

Comment 6: Consider adding a summary table that consolidates TF families predicted by both PlantRegMap and FootprintDB for easier comparison.

Response 6: We thank the reviewer for this valuable suggestion. To facilitate a more direct comparison as recommended, we have now added a new supplementary table (Table S5). This table consolidates the transcription factor families predicted by both PlantRegMap and FootprintDB, presenting a clear and comprehensive summary that allows for an easier and more effective comparison between the two datasets.

Comment 7: The use of both PlantRegMap and FootprintDB strengthens the analysis; however, the weak correlation should be discussed in more detail to help the reader understand possible reasons.

Response 7: We thank the reviewer for their valuable feedback. To address the comment regarding the use of PlantRegMap and FootprintDB and the resulting weak correlation, we have revised the manuscript to provide additional context and clarification. Specifically, we have added more detailed descriptions of the methodologies for PlantRegMap in Section 2.1 and for FootprintDB in Section 2.3. Furthermore, in Section 2.4, we have expanded the discussion to better explain why the weak correlation is an expected and informative outcome, highlighting the complementary nature of these two distinct bioinformatic approaches. We believe these additions strengthen the manuscript and provide greater clarity for the reader.

Comment 8: The repeated emphasis on AP2/ERF and BBR-BPC families is good, but highlighting novel/unexpected TF families (like FRS or Dof) could make the results more balanced.

Response 8: We appreciate the reviewer's insightful comment regarding the need for a more balanced discussion of the identified transcription factor (TF) families. We agree that highlighting novel findings related to the Dof and FRS families enhances the manuscript. To address this, we have revised both the Results and Discussion sections. In the Results (lines 197-203), we now explicitly mention the identification of TFs from these less common families and discuss how they contribute to a multi-layered regulatory network. Furthermore, we have updated the Discussion by changing the heading of Section 3.1 and adding a new paragraph (lines 299-308) that is dedicated to exploring the potential significance of the Dof and FRS families, referencing their known roles and the novel complexity they introduce to the regulation of PAL genes. We believe these changes effectively incorporate the reviewer's suggestion and provide a more comprehensive interpretation of our findings.

Comment 9: “This finding contributes to our understanding of the complex regulatory mechanisms…” should be “contributes.”

Response 9: Thank you for the feedback. The reviewer is correct; the subject of the sentence, "This finding," is singular, and therefore the verb should be "contributes" to ensure proper subject-verb agreement. The sentence has been corrected in the revised manuscript to reflect this change.

Comment 10: “The analysis provides detailed information, including the exact matching sequence…” should be “provides.”

Response 10: Thank you for your valuable feedback and careful review of our manuscript. We appreciate you pointing out this grammatical error. We have corrected the sentence in the manuscript to "The analysis provides detailed information..." as you suggested, ensuring the text is accurate and clear.

Comment 11: In section 3.2, the heading says “NAC083 plays as activator” — revise to “NAC083 acts as an activator” for grammatical accuracy and clarity.

Response 11: We appreciate the reviewer's suggestion to improve the grammatical accuracy of our manuscript. In accordance with your comment, we have revised the heading in Section 3.2 from “NAC083 plays as activator” to “NAC083 acts as an activator” to enhance clarity and precision. Thank you for this valuable correction.

Comment 12: It effectively combines both the biological process and experimental condition, making it informative and scientifically strong.

Response 12: We thank the reviewer for their positive feedback. We are pleased that they found the integration of the biological process and the experimental condition to be effective and scientifically strong, as this was a key objective of our study. We appreciate the encouraging comment.

Comment 13: Many grammatical and language mistakes should be improved.

Response 13: Thank you for your constructive feedback. We have thoroughly proofread the entire manuscript and corrected the grammatical errors and improved the language throughout the text. We believe the revisions have significantly enhanced the clarity and readability of our work.

Comment 14: Please avoid symbols at the beginning of sentences.

Response 14: We appreciate the reviewer's constructive feedback. We have carefully reviewed the entire manuscript and have revised all sentences to ensure they do not begin with a symbol.

Comment 15: Please arrange all the references according to the Journal format.

Response 15: We thank the reviewer for their valuable feedback. In response to the comment, we have carefully revised the entire reference section to fully comply with the journal's formatting guidelines. Specifically, all journal names have now been updated to their standard abbreviated format as requested.

Reviewer 2 Report

Comments and Suggestions for Authors

Authors used the bioinformatic tools to characterize cis-active motifs and corresponding putative trans-factors in promoters of six phenylpropanoid pathway genes in mulberry. Authors identified three statistically significant conserved motifs within the 2000 bp promoter  region. Then, the TFs binding sites were predicted with FootprintDB and compared with results for PlantRegMap . Authors observed a weak correlation between both databases. Transcriptomic analysis was used to identify trans-factors associated with the response to chitosan treatment.

Study provides novel and important informations that could be interesting to researchers in the field. Study is well planned and performed. However, the quality of Figures and the description of presented methods should be improved. Following comments should be addressed:

  1. Line 139-remove the excessive dot.
  2. Line 182 Correct the sentence; instead of number [19] use Rakpenthai et al. (2022)
  3. Fig 3b It is hard to see which particular cis-active motif is common or not among these loci. Maybe use a supplementary table to clearly show these differences.
  4. Fig 4 It is not possibile to read names of trans-factors, paricularly below the red line.
  5. Fig. 5. Values Log2 FC on Fig. 5 differ from those presented in table S3, in the last column AT. Explain and correct it.
  6. Results- present the number or raw and clean reads for transcriptomic experiment in a separate table.
  7. Section 4.3 Add information related to photoperiod or light intensity.
  8. Section 4.4 How the quality of RNA was checked and assured? Add the volume and concentration of libraries.
  9. Authors could add a few sentences in the Discussion section, to show that TF activity depends on position, binding-site strand, chromatin context, or cooperative interaction with other TFs. You may use these citations or related:

de Boer CG, Vaishnav ED, Sadeh R, Abeyta EL, Friedman N, Regev A. Deciphering eukaryotic gene-regulatory logic with 100 million random promoters. Nat Biotechnol. 2020 Jan;38(1):56-65. doi: 10.1038/s41587-019-0315-8. Epub 2019 Dec 2. Erratum in: Nat Biotechnol. 2020 Oct;38(10):1211. doi: 10.1038/s41587-020-0665-2.

 Kim, Seungsoo et al. Deciphering the multi-scale, quantitative cis-regulatory code. Molecular Cell, Volume 83, Issue 3, 373 – 392.

Author Response

Comment 1: Line 139-remove the excessive dot.

Response 1: Thank you for your careful attention to detail. We have corrected the manuscript by removing the excessive dot at line 139 as you suggested.

Comment 2: Line 182 Correct the sentence; instead of number [19] use Rakpenthai et al. (2022)

Response 2: Thank you for the suggestion. We have revised the manuscript as requested. The numerical citation [19] on line 182 has been corrected to the text citation format, "Rakpenthai et al. (2022)," to adhere to the proper citation style.

Comment 3: Fig 3b It is hard to see which particular cis-active motif is common or not among these loci. Maybe use a supplementary table to clearly show these differences.

Response 3: We thank the reviewer for this valuable suggestion to enhance the clarity of our findings. To address this concern, we have now created a new supplementary table (Table S4) which summarizes the data presented in the Venn diagram (Figure 3b). This table clearly lists the specific cis-active motifs that are common and unique among the different gene loci, providing a more detailed and easily accessible breakdown of their distribution.

Comment 4: Fig 4 It is not possibile to read names of trans-factors, paricularly below the red line.

Response 4: We appreciate the reviewer's valuable feedback regarding the readability of the labels in Figure 4. We have now revised the figure to correct this issue. The positions of the transcription factor family names have been adjusted to prevent any overlap, ensuring that every name is now clearly visible and legible for easier interpretation. The updated Figure 4 has been incorporated into the revised manuscript.

Comment 5: Fig. 5. Values Log2 FC on Fig. 5 differ from those presented in table S3, in the last column AT. Explain and correct it.

Response 5: We thank the reviewer for their careful observation and wish to clarify the contents of the supplementary table. The values plotted in Figure 5 are the average Log₂ Fold Change (Log₂FC), which are correctly located in column AG ('LOG2_CHIvsFN') of Table S3. The column the reviewer refers to, column AT ('SEM_Log2FC'), contains the standard error of the mean (SEM) associated with those Log₂FC values, not the primary values themselves. To prevent future ambiguity, we have added a remark within Table S3 to explicitly guide readers to the correct columns for both the Log₂FC and SEM data corresponding to Figure 5.

Comment 6: Results- present the number or raw and clean reads for transcriptomic experiment in a separate table.

Response 6: Thank you for the suggestion. We would like to clarify that detailed information regarding the number of raw and clean reads for the transcriptomic experiment is available within the raw sequencing data. As stated in the Data Availability Statement, the raw sequencing data from this study is publicly accessible through the NCBI Gene Expression Omnibus (GEO) under the accession number GSE303913.

Comment 7: Section 4.3 Add information related to photoperiod or light intensity.

Response 7: Thank you for your valuable suggestion. We have revised the manuscript and added the specific details regarding the photoperiod condition used for plant cultivation in Section 4.3, as requested. We believe this additional information provides a clearer and more complete description of our experimental setup.

Comment 8: Section 4.4 How the quality of RNA was checked and assured? Add the volume and concentration of libraries.

Response 8: We thank the reviewer for their valuable feedback. To address this concern, we have revised the manuscript to provide the requested details. We have now specified in Section 4.3 that the quality of the extracted RNA was assured using FastQC analysis to assess the Phred quality score. Additionally, in Section 4.4, we have clarified that the final library concentrations were quantified using qPCR and were required to be at least 10 nM for successful sequencing on the Illumina HiSeq platform.

Comment 9: Authors could add a few sentences in the Discussion section, to show that TF activity depends on position, binding-site strand, chromatin context, or cooperative interaction with other TFs. You may use these citations or related:

de Boer CG, Vaishnav ED, Sadeh R, Abeyta EL, Friedman N, Regev A. Deciphering eukaryotic gene-regulatory logic with 100 million random promoters. Nat Biotechnol. 2020 Jan;38(1):56-65. doi: 10.1038/s41587-019-0315-8. Epub 2019 Dec 2. Erratum in: Nat Biotechnol. 2020 Oct;38(10):1211. doi: 10.1038/s41587-020-0665-2.

 Kim, Seungsoo et al. Deciphering the multi-scale, quantitative cis-regulatory code. Molecular Cell, Volume 83, Issue 3, 373 – 392.

Response 9: Thank you for this valuable suggestion. We agree that acknowledging the complex nature of transcription factor (TF) regulation is crucial. In response, we have revised the manuscript by adding sentences to the end of Discussion section 3.1. This addition now clarifies that the predictive binding of TFs is only the first step, and their actual regulatory activity in vivo is influenced by a multitude of factors, including the precise position of their binding sites, the local chromatin context, and potential cooperative or competitive interactions with other TFs. We believe this clarification adds important context to our findings.

Reviewer 3 Report

Comments and Suggestions for Authors

This manuscript presents an integrated analysis of transcriptional regulation of the Phenylalanine Ammonia-Lyase (PAL) gene family in mulberry (Morus notabilis) under chitosan-induced stress. The authors combine in silico promoter motif analysis (MEME suite, PlantRegMap, FootprintDB) with transcriptomic profiling to identify key transcription factors (TFs) potentially regulating PAL expression. They propose NAC083 as an activator and members of AP2/ERF and BBR-BPC families as repressors, providing a valuable framework for understanding phenylpropanoid pathway regulation.

The work is relevant and well-structured. The combination of computational prediction with transcriptomic validation adds robustness to the findings, and the study has potential application in mulberry improvement and secondary metabolite engineering.

Minor revisions recommended

These points are largely about clarity, consistency, and transparency rather than changing the core results.

Consistency in gene IDs and locus names: some locus IDs appear with inconsistent numbering (e.g., LOC24107112 vs LOC21407112). Please ensure they match the genome annotation and are consistent throughout.

Clarification of promoter region definition: you mention both 2000 bp upstream and core promoter 200 bp regions. It would help to clearly state which region was analyzed for each tool (PlantRegMap, MEME, FootprintDB) and why.

Method details for reproducibility: please provide the exact parameters used in MEME (min/max motif width, number of motifs, distribution model). Also, clarify which Morus genome annotation was used for promoter extraction and how the transcription start site (TSS) was defined.

Wording adjustments for interpretation: while the transcriptome data strongly support your hypotheses, please slightly soften statements such as “confirming” or “validating” TF roles, as functional binding assays would be needed for confirmation. Example: change “confirming NAC083 as an activator” to “supporting NAC083 as a putative activator.”

Experimental details for chitosan treatment: please, indicate the final concentration in % (w/v) and whether pH was adjusted after dissolution, as these can influence plant responses.

Figures: please improve the quality of the figures.

Overall recommendation

The manuscript is well-written, the study is sound, and the results are of interest to the plant molecular biology community. The suggested changes are straightforward and aimed at improving clarity and reproducibility rather than altering the main findings.

This work will make a valuable contribution to our understanding of phenylpropanoid pathway regulation in mulberry and sets the stage for future functional characterization of the identified transcription factors.

Author Response

Comment 1: Consistency in gene IDs and locus names: some locus IDs appear with inconsistent numbering (e.g., LOC24107112 vs LOC21407112). Please ensure they match the genome annotation and are consistent throughout.

Response 1: We sincerely thank the reviewer for their careful attention to detail and for identifying this inconsistency. We have corrected the typographical error in the manuscript and can confirm that the correct locus ID is LOC21407112. Furthermore, we have now thoroughly reviewed the entire manuscript to ensure that all gene IDs and locus names are consistent and correctly match the official genome annotation.

Comment 2: Clarification of promoter region definition: you mention both 2000 bp upstream and core promoter 200 bp regions. It would help to clearly state which region was analyzed for each tool (PlantRegMap, MEME, FootprintDB) and why.

Response 2: We thank the reviewer for this valuable feedback. We have revised the manuscript to clearly state which promoter region was analyzed by each tool and the rationale behind the selection. For the discovery of novel regulatory motifs, including potential distal elements, the MEME suite was applied to the full 2000 bp upstream sequence. In contrast, PlantRegMap analysis was focused on the 200 bp core promoter region to identify known transcription factor binding sites within this highly conserved area. Finally, FootprintDB was used to perform a high-confidence, targeted analysis specifically on the statistically significant motifs previously identified by MEME. These clarifications have now been added to sections 4.1 and 4.2 of the Materials and Methods.

Comment 3: Method details for reproducibility: please provide the exact parameters used in MEME (min/max motif width, number of motifs, distribution model). Also, clarify which Morus genome annotation was used for promoter extraction and how the transcription start site (TSS) was defined.

Response 3: We thank the reviewer for their constructive feedback. As detailed in the Methods section, the parameters for the MEME suite analysis were configured to identify up to 20 distinct motifs, each ranging from 16 to 18 nucleotides in length, which is typical for transcription factor binding sites. To ensure statistical significance, these motifs were filtered using an E-value threshold of ≤0.05. The manuscript also specifies that the promoter sequences were obtained from the Morus notabilis genome annotation available in the KEGG database (https://www.kegg.jp/kegg-bin/show_organism?org=mnt). We chose to define the upstream sequence relative to the translation start site (ATG) rather than the transcription start site (TSS) because the region between the TSS (+1) and the ATG codon can also contain functional regulatory regions to which transcription factors can bind.

Comment 4: Wording adjustments for interpretation: while the transcriptome data strongly support your hypotheses, please slightly soften statements such as “confirming” or “validating” TF roles, as functional binding assays would be needed for confirmation. Example: change “confirming NAC083 as an activator” to “supporting NAC083 as a putative activator.”

Response 4: We appreciate the reviewer's insightful comment and fully agree that the language should be moderated to accurately reflect the scope of our study. As our findings are based on transcriptomic and in silico evidence rather than functional binding assays, we have revised the manuscript to frame our conclusions as strongly supported hypotheses. We have carefully replaced definitive terms like "confirming" and "validating" with more cautious phrasing such as "supporting" and "suggesting" throughout the Abstract, Results, and Discussion sections. For example, we now describe NAC083 as a "putative activator" and a "strong candidate" to ensure our interpretations are precisely aligned with the experimental evidence presented.

Comment 5: Experimental details for chitosan treatment: please, indicate the final concentration in % (w/v) and whether pH was adjusted after dissolution, as these can influence plant responses.

Response 5: We thank the reviewer for their valuable suggestion. As requested, we have revised the manuscript to include the precise details of the chitosan treatment. The text now specifies that the final concentration of the medium molecular weight chitosan used was 0.04% (w/v) and clarifies that the pH of the medium was adjusted to 6.5 with NaOH after the chitosan was dissolved. We believe this addition provides the necessary clarity regarding our experimental protocol.

Comment 6: Figures, please improve the quality of the figures.

Response 6: We appreciate the reviewer's feedback regarding the figures. All figures in the manuscript have been carefully revised to enhance their quality and clarity. We have increased the resolution of each figure to ensure sharpness, improved the labeling for better readability, and adjusted the layouts for a more professional presentation. We are confident that the updated figures now meet the standards required for publication.

Round 2

Reviewer 2 Report

Comments and Suggestions for Authors

Authors corrected the manuscript according to suggestions, Manuscript in now significantly improved, I have no other comments.

Author Response

Comment 1: Open Review

Response 1: Thank you for the update regarding the reviews for our manuscript. We would like to inform you that we fully respect the decision of Reviewer #2 to not sign their review report for Open Review. We are completely comfortable with their review remaining anonymous and are grateful for their valuable feedback.

Comment 2: Authors corrected the manuscript according to suggestions, Manuscript in now significantly improved, I have no other comments.

Response 2: We sincerely thank the reviewer for their positive feedback and for recognizing the significant improvements in our manuscript. We are grateful for their time and constructive comments throughout the review process, which have been invaluable in strengthening our work.